# MODEL-AGNOSTIC SAMPLE REWEIGHTING FOR RELIABLE STRENGTH BEHAVIOR PREDICTION OF COARSE-GRAINED SOILS UNDER DISTRIBUTION SHIFTS

## ABSTRACT

Coarse-grained soils are widely employed in geotechnical engineering due to their favorable compaction and load-bearing properties. However, accurately modeling their strength behavior, typically characterized by the deviatoric stress–axial strain $(q - \varepsilon_a)$ relationship, remains challenging under real-world conditions, particularly when training and testing data exhibit distribution shifts. Existing deep learning approaches often rely on the independent and identically distributed (i.i.d.) assumption, limiting their robustness and generalization to out-of-distribution (OOD) scenarios. To address this limitation, we propose Model-Agnostic Sample Reweighting (MASR), a novel framework that integrates sample reweighting and invariant learning via bilevel optimization. In this approach, the inner loop optimizes the model on weighted samples, while the outer loop adjusts these weights to enhance generalization under OOD conditions. This iterative mechanism enables the model to progressively identify invariant and causally relevant features while suppressing spurious correlations. Empirical evaluations on real-world coarse-grained soil datasets demonstrate that MASR outperforms baseline models in predictive accuracy under significant distribution shifts.

## 1 INTRODUCTION

Coarse-grained soils are defined as mixtures with more than 50% of particles larger than 0.075 mm (Liu & Shi, 2019). Thanks to their good compaction and load-bearing performance, they are widely applied in engineering works such as dams, railways, highways, and bridge foundations (Shi Zhenming & Ming, 2014). However, due to their discontinuous, heterogeneous, and anisotropic nature, their mechanical behavior is more complex than that of finer soils. Understanding their strength behavior is therefore essential to ensure safety in design and construction.

The rapid progress of deep learning in recent years has provided novel avenues for investigating the strength behavior of coarse-grained soils. Owing to their ability to model complex nonlinear patterns, deep learning approaches have been actively explored in this domain (Verma & Kumar, 2022; Isik & Ozden, 2013), and have demonstrated strong predictive capabilities when trained on large-scale datasets. Nevertheless, several key challenges remain, with out-of-distribution (OOD) generalization being particularly critical. This issue stems from the standard assumption that data samples are independent and identically distributed (i.i.d.), which is frequently violated in real-world settings (Beery et al., 2018). In this study, such violations are primarily caused by inherent variations in properties like particle size distribution (PSD) and mineral composition across soil samples (Verma & Kumar, 2022), making OOD generalization a central challenge in modeling the strength behavior of coarse-grained soils under realistic conditions.

Enabling deep models to maintain generalization under distribution shifts remains a pressing and unresolved challenge. To tackle this, prior research has mainly progressed along two directions: one line of work focuses on applying regularization during training, while the other centers on reweighting training samples. Regularization-based methods include frameworks such as Distributionally Robust Optimization (DRO)(Ben-Tal et al., 2013) and Invariant Risk Minimization (IRM)(Arjovsky et al., 2019). IRM aims to suppress spurious correlations by learning invariant features, whereas DRO seeks to ensure performance under worst-case perturbations around the training distribution.

Although both have shown promise on small-scale datasets and shallow networks, their effectiveness declines notably when extended to deep architectures (Lin et al., 2021). This is largely due to over-parameterization: deep models can easily minimize the regularization objective to near-zero while still relying on unstable, spurious patterns, thereby compromising OOD generalization. Alternatively, sample reweighting methods, such as Importance Sampling (Ben-Tal et al., 2013) and Stable Learning (Kuang et al., 2020), aim to reshape the data distribution to reduce the model's reliance on spurious features. Importance Sampling adjusts sample weights inversely to group prevalence, thereby emphasizing minority instances, while Stable Learning reweights samples to promote orthogonality across features, reducing interference from irrelevant signals. Despite their theoretical appeal, they often require strong prior knowledge, such as explicit group annotations, which constrains their applicability in practice. To address these limitations, we propose a novel approach that combines the advantages of both regularization and reweighting strategies while overcoming their respective weaknesses.

We introduce a novel approach, Model-Agnostic Sample Reweighting (MASR), which reframes the conventional empirical risk minimization (ERM) process by transferring the optimization objective from model parameters to sample weights. This shift helps mitigate the overfitting issues frequently encountered in regularization-based methods. Moreover, by automatically learning the importance of each sample, MASR reduces the dependency on prior knowledge that typically constrains traditional reweighting strategies. MASR is built upon a bilevel optimization framework tailored to investigate the strength behavior of coarse-grained soils. In the inner loop, a neural network is trained on the weighted training samples using fixed sample weights. The resulting model is then assessed in the outer loop using out-of-distribution (OOD) criteria to evaluate its reliance on spurious correlations. The two loops interact iteratively, with the outer loop guiding the adjustment of sample weights in the inner loop by minimizing the OOD-based objective. By focusing the optimization on sample weights, MASR produces a weighting scheme that enhances the model's generalization under distribution shifts. These weights are model-agnostic and can be effectively transferred to other models addressing the same task.

The main contributions of this work are as follows:

**1. Robust Prediction under Distribution Shifts:** We propose MASR (Model-Agnostic Sample Reweighting), a new method designed to predict the strength behavior of coarse-grained soils in the presence of data distribution shifts—an issue frequently encountered in real-world geotechnical scenarios. By leveraging bilevel optimization to automatically learn sample weights, MASR enhances out-of-distribution (OOD) generalization and ensures greater robustness in prediction.

**2. Extending Invariant Learning to Regression:** This work frames the modeling of the deviatoric stress–axial strain $(q-\varepsilon_a)$ relationship as a regression task and applies invariant learning in this context. While most prior work on invariant learning (e.g., IRM) focuses on classification, we extend its application to regression for the first time in this domain, offering a novel approach for geotechnical strength analysis.

## 2 RELATED WORK

With the rise of artificial intelligence, deep learning has become an increasingly important tool in geotechnical engineering. As early as 1991, J. Ghaboussi (Ghaboussi et al., 1991) pioneered the use of neural networks to model material behavior, focusing on the stress–strain response of sandy soils. Later, Penumadu (Penumadu & Zhao, 1999) applied a feedback neural network to capture both stress–strain and volumetric changes in sandy soils, emphasizing the role of strain history in modeling the nonlinear behavior of coarse-grained soils. Kohestani (Kohestani & Hassanlourad, 2016) further explored this direction using both artificial neural networks and support vector machines to simulate the mechanical response of carbonate sands, demonstrating the predictive strength of neural networks for nonlinear soil behavior. Despite these advancements, deep learning methods remain fundamentally data-driven. Their performance often deteriorates under distribution shifts, which are common in real-world scenarios. Moreover, their black-box nature limits interpretability, posing significant challenges for practical engineering applications.

To enhance the out-of-distribution (OOD) generalization of deep learning models, two main strategies have been explored: sample reweighting and regularization-based approaches. Sample reweight-

ing, a classical technique for correcting distributional bias, typically assumes access to the test distribution and adjusts the influence of training instances to better match it, thereby enhancing generalization (Sugiyama et al., 2007; Fang et al., 2020). However, this assumption rarely holds in practical applications, where the test distribution is often unknown (Liu et al., 2021). To address this, methods such as Stable Learning have been proposed, which reassign sample weights to maintain statistical independence among features in the reweighted space (Shen et al., 2020; Wang et al., 2022). Despite their theoretical appeal, such approaches often rely on explicitly defined feature sets, which limits their applicability in complex real-world tasks.

On the other hand, regularization-based techniques, such as Invariant Risk Minimization (IRM), aim to incorporate feature invariance into the learning process (Peters et al., 2016). Various enhancements have been developed to improve IRM's performance. For instance, Krueger et al.(Krueger et al., 2021) and Xie et al.(Xie et al., 2020) introduced penalties based on risk variance across environments, while Chang et al.(Chang et al., 2020) and Xu et al.(Xu et al., 2021) proposed neural network-based methods to estimate and control the degree of invariance violation. In addition to invariance-based methods, Distributionally Robust Optimization (DRO) minimizes the worst-case loss over plausible data distributions to account for shift during training (Rothenhäusler et al., 2021). Nonetheless, both IRM and DRO often fail to perform well when applied to over-parameterized deep neural networks, as they are prone to overfitting (Gulrajani & Lopez-Paz, 2020; Lin et al., 2022).

## 3 METHODOLOGY

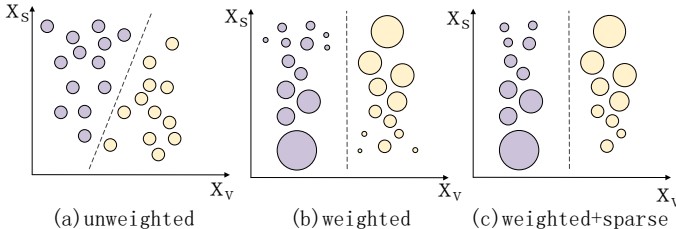

Figure 1: Eliminating spurious relationships through sample reweighting

In this study, we combine Invariant Risk Minimization (IRM) with sample reweighting to better capture causal patterns. As shown in Figure 1, IRM is first used to identify invariant relationships, followed by reweighting to reduce the model's reliance on spurious features. In the figure, each circle denotes a sample, with colors indicating class labels. $X_v$ represents invariant features, and $X_s$ refers to spurious features that vary across environments. Figure 1(a) shows a standard model trained without reweighting, where both $X_v$ and $X_s$ influence predictions. In contrast, Figure 1(b) depicts the outcome after reweighting, where larger circles indicate samples with greater weights, emphasizing stable correlations between $X_v$ and the label $Y$. The predictions thus align more closely with invariant features. To further reduce noise, Figure 1(c) introduces a sparsity constraint, which downweights uninformative samples, improving robustness and computational efficiency.

In the case of coarse-grained soils, strength behavior is affected by a range of factors such as test conditions, soil properties, and particle composition, which may be entangled with confounders or influenced by data selection bias. To address these challenges, the proposed MASR framework integrates IRM and sample reweighting within a bilevel optimization structure. A detailed description of MASR follows in the next subsection.

### 3.1 THE FRAMEWORK OF MASR

MASR is formulated as a bilevel framework, as illustrated in Figure 2, consisting of an inner loop and an outer loop. The inner loop contains an Empirical Risk Minimization (ERM) module, which utilizes an artificial neural network (ANN) trained with a weighted loss function. The outer loop includes two auxiliary modules: Module 1 replicates the model parameters $\theta$ from the ERM module to assess its reliance on invariant features using an OOD criterion computed over the training data($D_{\mathrm{tr}}$), while Module 2 updates sample weights $\mathbf{w}$ and binary selection masks $\mathbf{m}$ to minimize

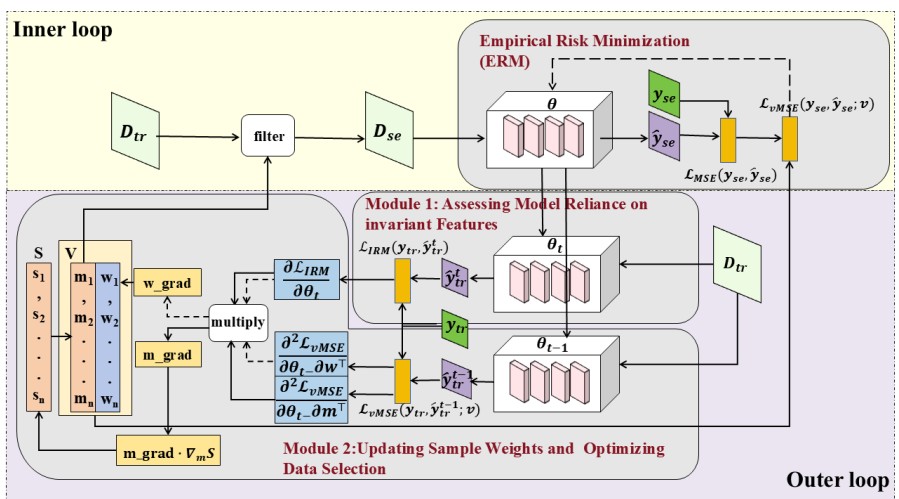

Figure 2: The architecture of the proposed MASR

this criterion. These weights $\mathbf{w}$ and masks $\mathbf{m}$ are then passed to the inner loop, where $\mathbf{m}$ is used to extract a selected subset to extract a subset $D_{\text{se}}$ from the full training data $D_{\text{tr}}$ and the masked sample weights $\mathbf{v}$ are computed accordingly ($\mathbf{v} = \mathbf{m} \odot \mathbf{w}$). The ERM module is subsequently trained on $D_{\text{se}}$ using $\mathbf{v}$. A more detailed architecture is shown in Figure 2, and the following subsections describe the full training process.

### 3.1.1 INNER LOOP

The inner loop is implemented as an Empirical Risk Minimization (ERM) module, where an artificial neural network (ANN) serves as the predictor. This module optimizes model parameters by minimizing a weighted loss function.

In line with the proposed sample reweighting scheme, each training instance is assigned a masked weight $v_i$, where $i$ indexes the samples. Let $x_i$ denote the input features and $y_i$ the corresponding ground truth label for the $i$-th sample. The model output is denoted by $\hat{y}_i$. A mean squared error (MSE) loss is computed for each sample and scaled by its weight $v_i$, resulting in a weighted loss function $\mathcal{L}_{\text{MSE}}(y_i, \hat{y}_i; \mathbf{v}, \boldsymbol{\theta})$, as defined in Equation 2, which is used to update the ANN parameters within the ERM module.

$$\ell(y_i, \hat{y}_i; \boldsymbol{\theta}) = (y_i - \hat{y}_i)^2 \tag{1}$$

$$\mathcal{L}_{\text{MSE}}(y_i, \hat{y}_i; \mathbf{v}, \boldsymbol{\theta}) = \frac{1}{n} \sum_{i=1}^{n} v_i (y_i - \hat{y}_i)^2 \tag{2}$$

here, $n$ represents the total number of training samples, and $\ell(y_i, \hat{y}_i)$ denotes the squared error between the prediction $\hat{y}_i$ and the true label $y_i$, as shown in Equation equation 1.

### 3.1.2 OUTER LOOP

The outer loop in MASR is designed to regulate the inner ERM process, thereby improving the model's robustness against shifts in data distribution between the training and test domains.

**(1)Module 1: Evaluating the model's reliance on invariant features**

Module 1 replicates the model parameters $\boldsymbol{\theta}$ from the ERM module and evaluates the ANN's reliance on invariant features by computing the out-of-distribution (OOD) criterion. This criterion is formulated based on the loss function of Invariant Risk Minimization (IRM), denoted as $\mathcal{L}_{\text{IRM}}$, as defined in Equations equation 3 and equation 4.

Invariant Risk Minimization (IRM) serves as a fundamental principle to evaluate the extent to which the trained ANN relies on invariant features. It assumes the existence of multiple environments $\mathcal{E} = \{e_1, e_2, \ldots, e_n\}$, where each environment $e_j$ is associated with a distinct joint distribution of features. Let $\mathbf{x}$ denote the original input features, which are composed of spurious features $\mathbf{x}_s$, prone to vary across environments, and invariant features $\mathbf{x}_v$, which remain stable. IRM formulates the prediction model as a composition of a feature extractor $g(\cdot; \boldsymbol{\Phi})$ and a predictor $h(\cdot; \boldsymbol{\psi})$. Thus, the overall prediction function is expressed as

$$f(\cdot; \boldsymbol{\theta}) = h(g(\cdot; \boldsymbol{\Phi}); \boldsymbol{\psi}),$$

where $\boldsymbol{\theta} = \{\boldsymbol{\Phi}, \boldsymbol{\psi}\}$ represents the complete set of model parameters.

The goal of IRM is to develop predictors that generalize to out-of-distribution (OOD) data, i.e., achieve low prediction error across multiple environments. To this end, IRM introduces a specialized loss function $\mathcal{L}_{\text{IRM}}$, which consists of an empirical risk term, and a regularization term that promotes feature invariance across environments. Two popular variants of this loss are $\mathcal{L}_{\text{IRMv1}}$ (Arjovsky et al., 2019) and$\mathcal{L}_{\text{REx}}$ (Krueger et al., 2021) , defined as follows:

$$\mathcal{L}_{\text{IRMv1}}(y, \hat{y}; \boldsymbol{\theta}) = \sum_{e \in \mathcal{E}} \mathcal{L}_{\text{MSE}}^e(y_i, \hat{y}_i; \boldsymbol{\theta}) \tag{3}$$
$$+ \lambda \left\| \nabla_{\boldsymbol{\psi}} \mathcal{L}_{\text{MSE}}^e(y_i, \hat{y}_i; \boldsymbol{\theta}) \right\|_2^2$$

$$\mathcal{L}_{\text{REX}}(y, \hat{y}; \boldsymbol{\theta}) = \sum_{e \in \mathcal{E}} \mathcal{L}_{\text{MSE}}^e(y, \hat{y}; \boldsymbol{\theta}) \tag{4}$$
$$+ \lambda \operatorname{Var}(\mathcal{L}_{\text{MSE}}^e(y, \hat{y}; \boldsymbol{\theta}))$$

In both equations, $y_i$ denotes the ground-truth labels, $\hat{y}_i$ represents the model predictions, and $\lambda$ is a hyperparameter balancing empirical risk and regularization.$\operatorname{Var}(\mathcal{L}_{\text{MSE}}^e(y, \hat{y}; \boldsymbol{\theta}))$ represents the variance of losses across different environments.

### (2)Module 2: Updating Sample weights and Optimizing data selection

The objective of Module 2 is to steer the ERM module toward greater reliance on invariant features while mitigating the influence of noisy samples, achieved by minimizing the OOD criterion computed by Module 1.

To this end, Module 2 introduces sample reweighting and the sparsity constraint. Each sample is assigned a weight $w_i$ to encourage the ERM module to place greater emphasis on invariant features. Additionally, a binary selection mask $m_i \in \{0, 1\}$ is applied to each sample to filter out noisy data. Both sample weights and selection masks are optimized by minimizing the OOD criterion.

### • Updating Sample weights

To compute the final sample weights ($w$), the gradient of the OOD criterion ($\mathcal{L}_{\text{IRM}}$) with respect to $w$ is utilized for updates. As $\mathcal{L}_{\text{IRM}}$ is not explicitly defined in terms of $w$, the chain rule is applied to derive the gradient, as shown below:

$$\nabla_w \mathcal{L}_{\text{IRM}} = \frac{\partial \mathcal{L}_{\text{IRM}}}{\partial \boldsymbol{\theta}} \cdot \frac{\partial \boldsymbol{\theta}}{\partial w} \approx \frac{\partial \mathcal{L}_{\text{IRM}}}{\partial \boldsymbol{\theta}_t} \cdot \frac{\partial \boldsymbol{\theta}_t}{\partial w} \tag{5}$$

where $\boldsymbol{\theta}$ denotes the model parameters obtained from the inner loop. Equation equation 5 approximates $\boldsymbol{\theta}$ by $\boldsymbol{\theta}_t$ obtained from $T$ steps of inner loop gradient descent, as illustrated in Figure 2.

The term $\frac{\partial \mathcal{L}_{\text{IRM}}}{\partial \boldsymbol{\theta}_t}$ can be directly computed according to Equations equation 3 or equation 4. However, $\frac{\partial \boldsymbol{\theta}_t}{\partial w}$ cannot be obtained from these equations, as $\boldsymbol{\theta}_t$ is not explicitly expressed with $w$. The computation of $\frac{\partial \boldsymbol{\theta}_t}{\partial w}$ forms a bilevel optimization problem. To tackle this problem, we adopt the

1-step Truncated Backpropagation method proposed by Shaban et al. (Shaban et al., 2019) to approximately compute the gradient of $\boldsymbol{\theta}_t$ with respect to $w$. The resulting gradient computation is described in Equationequation 6.

$$\frac{\partial \theta_T}{\partial \boldsymbol{w}} = \sum_{j \leq L} \left[ \prod_{k<j} \left( I - \frac{\partial^2 \mathcal{L}}{\partial \theta \partial \theta^\top} \Big|_{\theta_{T-k-1}} \right) \frac{\partial^2 \mathcal{L}}{\partial \theta \partial \boldsymbol{w}^\top} \Big|_{\theta_{T-j-1}} \right]$$
$$\approx \frac{\partial^2 \mathcal{L}_{\text{vMSE}}}{\partial \theta_{T-1} \partial \boldsymbol{w}^\top} \tag{6}$$

As shown in Equation equation 6, $L$ denotes the truncation length considered during backpropagation. The parameter $\theta_{t-1}$ is used to update $\boldsymbol{w}$, as illustrated in Module 2 of Figure 2.

Accordingly, the gradient of the OOD criterion ($\mathcal{L}_{IRM}$) with respect to $\mathbf{w}$, originally defined in Equation equation 5, is approximated using Equation equation 7. Correspondingly, the sample weights $\mathbf{w}$ are updated based on this approximation, as shown in Equation equation 8. Here, the learning rate $\eta$ controls the step size of the update, and Projected Gradient Descent (PGD) is employed to ensure that the sample weights remain positive. The operator $\text{proj}_C$ denotes PGD, which projects the updated $\mathbf{w}$ onto a constraint set $C$, where $C = \{\mathbf{w} : \mathbf{w} > 0\}$.

$$\nabla_{\mathbf{w}} \mathcal{L}_{IRM} \approx \frac{\partial \mathcal{L}_{IRM}}{\partial \theta_t} \cdot \frac{\partial^2 \mathcal{L}_{vMSE}}{\partial \theta_{t-1} \partial \mathbf{w}^\top} \tag{7}$$

$$\mathbf{w} \leftarrow \text{proj}_C \left( \mathbf{w} - \eta \frac{\partial \mathcal{L}_{IRM}}{\partial \theta_t} \cdot \frac{\partial^2 \mathcal{L}_{vMSE}}{\partial \theta_{t-1} \partial \mathbf{w}^\top} \right) \tag{8}$$

• **Optimizing data selection**

The gradient of the binary mask $m$ is computed using a strategy similar to that used for $\mathbf{w}$, as shown in Equation equation 9.

$$\nabla_{\mathbf{m}} \mathcal{L}_{IRM} \approx \frac{\partial \mathcal{L}_{IRM}}{\partial \theta_t} \cdot \frac{\partial^2 \mathcal{L}_{vMSE}}{\partial \theta_t \partial \mathbf{m}^T} \Big|_{\theta_{t-1}} \tag{9}$$

However, due to the discrete nature of $m$, it is not directly suitable for gradient-based optimization. Instead, we introduce a probability vector $\mathbf{s}$, as illustrated in Figure 2, where each element $s_i \in [0, 1]$ denotes the probability of selecting the $i$-th sample. We calculate the gradient to s by Straight-through Gumbel-softmax (Paulus et al., 2020). Specifically, following a similar procedure to the update of $\mathbf{w}$ in Equation equation 8, we first optimize $\mathbf{s}$, as shown in Equations equation 10 and equation 11.

$$\nabla_{\mathbf{s}} \mathcal{L}_{IRM} = \nabla_{\mathbf{m}} \mathcal{L}_{IRM} \cdot \nabla_{\mathbf{s}} \mathbf{m} \tag{10}$$

$$\mathbf{s} \leftarrow \text{proj}_C \left( \mathbf{s} - \eta \frac{\partial \mathcal{L}_{IRM}}{\partial \mathbf{s}} \cdot \frac{\partial^2 \mathcal{L}_{vMSE}}{\partial \theta_{t-1} \partial \mathbf{s}^\top} \right) \tag{11}$$

where $\text{proj}_C$ denotes the projection of the updated $\mathbf{s}$ onto a constraint set $C'$, $C' = \{\mathbf{s} : 0 \leq s \leq 1\}$. The learning rate $\eta$ controls the update step size. The updated $\mathbf{s}$ is then binarized to obtain the new selection mask $m$ according to Equation equation 12 according to Straight-through Gumbel-softmax.

$$m_i = 1 \left( \log \frac{s_i}{1 - s_i} + g_1 - g_0 \geq 0 \right) \tag{12}$$

Here, $g_0$ and $g_1$ are two independent random variables that follow the Gumbel distribution of $Gumbel(0, 1)$.

---

**Algorithm 1** Model-Agnostic Sample Reweighting Algorithm (MASR)

---

**Input:** The number of the selected samples $K$, Training set $D_{tr}$, Maximum number of iterations $R$, Maximum number of training epochs $T$

**Output:** Parameters $(\theta)$ of ANN

1: Initialize model parameters $\theta_0$
2: Initialize all sample weights $\{w_i=1|i=1...n\}$
3: Initialize selection probabilities $\{s_i = \frac{K}{|D_{tr}|}|i=1...n\}$
4: **repeat**
5:     Generate selection masks $m$ from $s$ by Equation equation 12
6:     Calculate the gradient $\nabla_s m$
7:     Filter the training set $D_{tr}$ and obtain $D_{se}$ according to $m$
8:     Compute the masked weights $v$ ($v = m \circ w$)
9:     **repeat**
10:       Use samples in $D_{se}$ for training to get $\hat{y}_{se}$
11:       Calculate the weighted MSE Loss $\mathcal{L}_{vMSE}(y_i, \hat{y}_i; v; \theta)$ by Equation equation 2
12:       Update model parameters via gradient descent
13:     **until** the maximum number of training epochs $T$ is reached
14:     Copy the parameters $\theta_t$ obtained in the $T$-th epoch from the ERM module to Module 1
15:     Evaluate model $\theta_t$ on the full training set $D_{tr}$ by computing $\mathcal{L}_{IRM}$ using Equation equation 3 or equation 4
16:     Copy the parameters $\theta_{t-1}$ obtained in the $(T-1)$-th epoch from the ERM module to Module 2
17:     Use previous step $\theta_{t-1}$ on the full training set $D_{tr}$ to predict $\hat{y}_{tr}$ and compute the weighted MSE loss $\mathcal{L}_{vMSE}(y_{tr}, \hat{y}_{tr}; v; \theta)$
18:     Compute gradients $\nabla_w \mathcal{L}_{IRM}$ and $\nabla_m \mathcal{L}_{IRM}$ according to Equation equation 7 and equation 9
19:     Compute $\nabla_s \mathcal{L}_{IRM} = \nabla_m \mathcal{L}_{IRM} \cdot \nabla_s m$ according to Equation equation 10
20:     Update $w$ and $s$ according to Equation equation 8 and equation 11
21:     **return** parameters $\theta_t$
22: **until** the maximum number of iterations $R$ is reached

---

## 4 EXPERIMENT

### 4.1 CONSTRUCTION OF DATASETS WITH DISTRIBUTIONAL SHIFTS

This study relies entirely on real-world data obtained from triaxial compression tests. The dataset comprises 545 test groups, each containing approximately 50 paired observations of deviatoric stress and axial strain $(q - \varepsilon_a)$, resulting in a total of 24,398 data points. Notably, the maximum particle size $(d_{max})$ varies considerably across the tests, ranging from 10 mm to 200 mm.

To assess the effectiveness of the proposed approach, two separate datasets are constructed for evaluation purposes, as detailed below.

**Dataset A (Group 1):** As presented in Table 1, Dataset A includes both training and testing subsets, with the distribution of maximum particle sizes $(d_{max})$ clearly specified. This dataset is constructed to evaluate the performance of MASR under conditions where the training and test data follow similar distributional characteristics.

Table 1: Data distribution for dataset A

| Dataset | | Maximum particle size ($d_{max}$) | | | Sum |
|---|---|---|---|---|---|
| | | <60 mm | 50 - 60 mm | >60 mm | |
| A | Training Set | 0 | 15680 | 0 | 15680 |
| | Test Set | 0 | 0 | 3034 | 3034 |

**Dataset B (Group 2):** As shown in Table 2, Dataset B includes one training set and two test sets, each exhibiting distinct distributions of maximum particle size $(d_{max})$. Test Set 1 consists of samples

with relatively larger particle sizes, while Test Set 2 contains samples with smaller ones. This dataset is designed to assess the model's generalization capability under significant distributional shifts.

Table 2: Data distribution for dataset B

| Dataset | | Maximum particle size ($d_{\max}$) | | | Sum |
|---|---|---|---|---|---|
| | | $<60\,\mathrm{mm}$ | 50 - 60 mm | $>60\,\mathrm{mm}$ | |
| | Training Set | 0 | 18714 | 0 | 18714 |
| B | Test Set 1 | 0 | 0 | 2577 | 2577 |
| | Test Set 2 | 3107 | 0 | 0 | 3107 |

## 4.2 BASELINE METHODS

To validate the effectiveness of the proposed MASR, we compare it against three baseline models:

**ANN:** This study employs a conventional fully connected artificial neural network (ANN) as a baseline model. The hyperparameters are consistent with those used in MASR, which are provided in Table 7 in the appendix.

**SNN:** The Stable Neural Network (SNN) (Zhang et al., 2024) integrates a stable learning module into the ANN structure to improve generalization under distributional shifts. It is included as a direct benchmark for comparison with MASR.

**IRM:** Since MASR builds upon Invariant Risk Minimization (IRM) by introducing sample reweighting, the original IRM method is selected as a baseline to isolate the contribution of this enhancement. For fairness, IRM adopts the same ANN architecture as used in SNN and MASR.

## 4.3 EXPERIMENTAL RESULTS AND ANALYSIS

### 4.3.1 THE PERFORMANCE OF PROPOSED MASR

A thorough evaluation was conducted to compare the proposed MASR with standard ANN, IRM, and SNN models using two experimental settings based on different datasets. The corresponding results are summarized in Tables 3–5.

The first set of experiments includes only Experiment I, which utilizes Dataset A (Group 1) to assess model performance under conditions where the training and test sets share similar data distributions. As shown in Table 3, MASR consistently achieves superior results across all evaluation metrics, outperforming ANN, SNN, and IRM.

Table 3: Performance Comparison on Dataset A (Group 1)

| Expt. | Model | OOD criterion | MSE | MAE | $R^2$ | MAPE |
|---|---|---|---|---|---|---|
| | ANN | - | 1.3e6 | 859.7 | 0.9012 | 0.563 |
| | SNN | - | 9.0e5 | 746.5 | 0.9323 | 0.353 |
| I | MASR | IRMv1 | 7.7e5 | 677.6 | 0.9379 | 0.325 |
| | | REx | **7.6e5** | **677.5** | **0.9382** | **0.322** |
| | IRM | IRMv1 | 2.0e6 | 1240.3 | 0.8316 | 0.857 |
| | | REx | 2.4e6 | 1137.9 | 0.8483 | 0.837 |

Dataset B (Group 2) was employed in the second set of experiments to evaluate the model's ability to generalize under substantial distributional shifts. Specifically, Experiment II assesses performance on Test Set 1, whereas Experiment III targets Test Set 2.

The outcomes of Experiment II, presented in Table 4, show that MASR consistently surpasses all baseline models across every evaluation metric. Notably, under the REx-based OOD criterion, MASR attains an $R^2$ score of 0.8799, marking a 5.62% improvement over SNN and significantly outperforming ANN and IRM. These results highlight MASR's strong generalization ability when

applied to coarse-grained soils with large particle sizes absent from the training data. Considering that such samples with larger particle sizes are relatively uncommon in real-world datasets, MASR's superior performance demonstrates its practical value in handling both data scarcity and distributional variation.

Table 5 presents the results of Experiment III, indicating that the Model-Agnostic Sample Reweighting (MASR) method achieves the highest predictive accuracy across all evaluation metrics for coarse-grained soils with small particle sizes that were not included in the training set. This result further demonstrates the robustness of MASR in effectively addressing distributional shifts.

Comparing the results in Tables 4 and 5 reveals a notable discrepancy in predictive performance: MASR's accuracy on large-particle samples (Table 4) is significantly lower than that on small-particle samples (Table 5). This performance gap can be attributed to differences in the particle size distribution (PSD), which describes the proportion of soil particles finer than a given diameter and is commonly represented as a cumulative curve. Specifically, the PSD curves associated with small particle sizes span a narrower range and are well represented in the training dataset, thereby enabling more accurate predictions. In contrast, the PSD curves for large-particle samples exhibit a much broader range, introducing feature patterns that are underrepresented or even absent in the training data, which in turn limits predictive accuracy under these conditions.

A comparison of the results from the first two groups of experiments shows that the IRM method based on the ANN network performs the worst. This suggests that directly updating network parameters using OOD criteria is not effective for our task. In contrast, MASR proves more effective by shifting the optimization focus from model parameters to sample weights.

Table 4: Performance Comparison on Dataset B with test set 1 (Group 2)

| Expt. | Model | OOD criterion | MSE | MAE | R² | MAPE |
|---|---|---|---|---|---|---|
| II | ANN | - | 3.3e6 | 1490.4 | 0.7636 | 1.690 |
| | SNN | - | 3.0e6 | 1311.5 | 0.8237 | 1.3340 |
| | MASR | IRMv1 | 1.9e6 | 1021.3 | 0.8742 | 0.773 |
| | | REx | **1.9e6** | **993.9** | **0.8799** | **0.742** |
| | IRM | IRMv1 | 5.9e6 | 1756.1 | 0.6355 | 2.080 |
| | | REx | 6.2e6 | 1834.0 | 0.6098 | 02.216 |

Table 5: Performance Comparison on Dataset B with test set 2 (Group 2)

| Expt. | Model | OOD criterion | MSE | MAE | R² | MAPE |
|---|---|---|---|---|---|---|
| III | ANN | - | 3.8e6 | 875.2 | 0.9370 | 0.498 |
| | SNN | - | 1.1e6 | 523.1 | 0.9751 | 0.399 |
| | MASR | IRMv1 | 7.5e5 | 464.9 | 0.9809 | 0.415 |
| | | REx | **7.5e5** | **465.3** | **0.9811** | **0.414** |
| | IRM | IRMv1 | 8.2e6 | 1397.4 | 0.8369 | 0.558 |
| | | REx | 7.5e6 | 1302.9 | 0.8554 | 0.552 |

## 5 CONCLUSION

This study introduces the Model-Agnostic Sample Reweighting (MASR) method to improve prediction accuracy and model interpretability for predicting the strength behavior of coarse-grained soils under data distribution shifts. MASR addresses out-of-distribution (OOD) generalization through a bilevel optimization framework that integrates invariant learning with sample reweighting. In this framework, the model is trained in an inner loop with predefined sample weights, while the outer loop uses OOD criteria to refine the model's focus on causal relationships. A sparsity constraint is also incorporated to improve robustness and computational efficiency. Experimental results show that MASR consistently outperforms traditional neural networks (ANN) SNN and IRM, particularly in OOD scenarios, confirming its effectiveness in addressing practical engineering challenges.

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

# A APPENDIX

This appendix can be divided into the following parts:

1. Section A gives the details of the dataset

2. Section B presents the Model Configuration.

## A.1 A. DATASET DETAILS

### A.1.1 FUNDAMENTAL DEFINITIONS

The strength behavior of coarse-grained soils is influenced by several factors, which can be classified into three categories: **test conditions**, **soil state**, and **particle composition**.

In triaxial compression tests, **test conditions** refer to the specific environmental, mechanical, and procedural parameters controlled during the test to accurately simulate real-world soil behavior. This study focuses on two key test conditions: *Confining Pressure* and *Specimen Dimensions*.

*Confining Pressure* ($\sigma_3$): It is the pressure exerted on a soil specimen in the triaxial compression test by the surrounding confining material (typically water or air) that is applied in all directions. It simulates the in-situ lateral stress conditions and significantly affects the sample's deformation and shear strength behavior.

*Specimen Dimensions*: These are the geometric parameters of the cylindrical soil specimen used in triaxial compression tests, typically characterized by its height ($h$) and diameter ($d$), which can influence stress distribution and failure behavior.

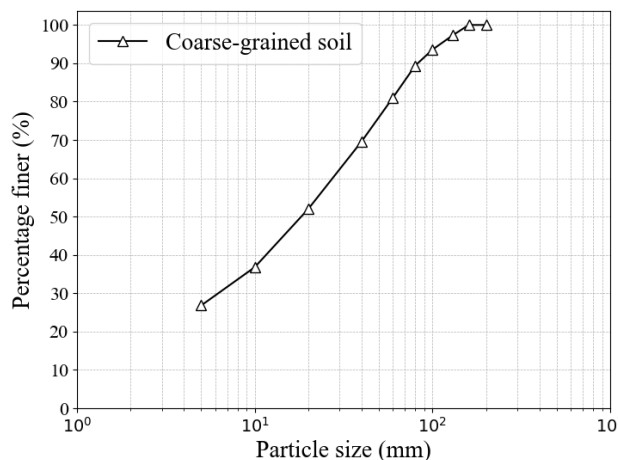

Figure 3: The discretization result of the particle size distribution (PSD) curve

**Soil state** refers to the condition or characteristics of the soil at the time of testing, which can significantly affect the strength behavior during mechanical testing, such as the triaxial compression test. Soil state typically includes factors such as:

*Dry Density* ($\rho_d$): It represents the mass of soil per unit volume when all the water content has been removed. It is an indicator of how compact the soil is when it contains no moisture. This value is used to determine the degree of compaction in soils for construction purposes. *Void Ratio* ($e$): The void ratio is a measure of the void spaces (pores or air gaps) in soil relative to the solid material. It is a crucial property in geotechnical engineering, used to describe soil density and compaction. The void ratio significantly affects soil strength, compressibility, and permeability. It is inversely related to dry density, as illustrated in Equations 13 and 14.

$$\rho_d = \frac{G_s \rho_w}{1 + e} \tag{13}$$

$$e = \left( \frac{G_s \rho_w}{\rho_d} - 1 \right) \times 100\% \tag{14}$$

where $G_s$ is the specific gravity of soil particles and $\rho_w$ is the density of water.

**Particle composition** refers to the makeup of the individual particles that constitute a soil sample. It describes the types, sizes, shapes, and distribution of mineral particles, organic material, and other components within the soil. In geotechnical engineering, *particle size distribution* (PSD) is a crucial indicator of particle composition, as it directly affects the soil's mechanical behavior.

*Particle Size Distribution* (PSD): It describes the range and proportion of particle sizes within a soil sample. It is a fundamental property in geotechnical engineering that describes how the individual particles in the soil are distributed across various sizes. PSD is typically represented as a curve, showing the percentage of particles passing through various sieve sizes. The horizontal axis represents the particle size (mm), while the vertical axis shows the cumulative passing percentage (%), indicating the proportion of particles that are finer than a given size by mass. PSD curves provide critical insights into soil gradation and are widely used to assess the strength behavior of coarse-grained soils. However, to make the continuous PSD curves suitable for predictive model inputs, they need to be discretized. In this study, the curves are discretized by extracting the cumulative percentages at predefined particle size breakpoints: 5 mm, 10 mm, 20 mm, 40 mm, 60 mm, 80 mm, 100 mm, 130 mm, 160 mm, and 200 mm. These discretized values effectively capture the shape of the PSD curve and serve as structured input features for the predictive model. Figure 3 illustrates a representative PSD curve for a material with a maximum particle size of 160 mm, discretized using the specified breakpoints.

### A.1.2 DATA ACQUISITION AND PREPROCESSING

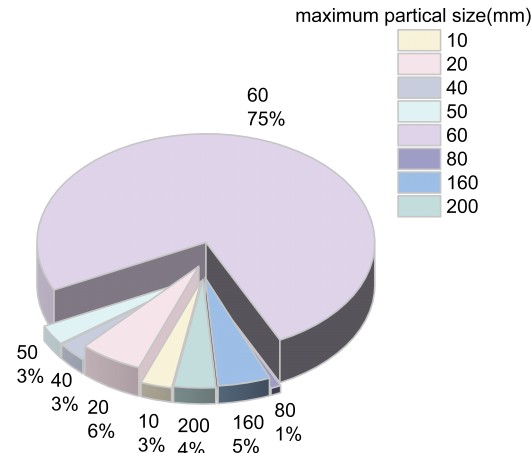

Figure 4: Distribution of Data Samples Across Different Maximum Particle Sizes

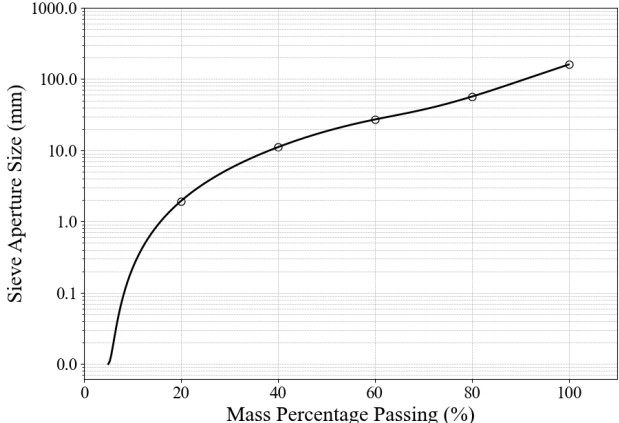

Figure 5: The PSD Curve Discretization Based on Mass Percentage Passing Through Predefined Sieve Apertures

Due to the high cost of triaxial compression tests, many studies rely on discrete element analysis software like PFC to generate datasets. However, these datasets often differ from real-world data, making it challenging for models to accurately capture the underlying patterns observed in practical scenarios. To ensure authenticity, this study exclusively utilizes real data obtained from triaxial compression tests. The dataset primarily consists of experimental results from the China Institute of Water Resources and Hydropower Research, supplemented by triaxial compression test data from English and Japanese literature. The dataset comprises 545 groups of triaxial compression tests, each consisting of approximately 50 deviatoric stress–axial strain $(q - \varepsilon_a)$ data points, resulting in a total of 24,398 observations. Figure 6 shows the result for one of the groups, where the $x$-axis represents axial strain $(\varepsilon_a)$ and the $y$-axis represents deviatoric stress $(q)$.

Among the triaxial compression tests, the maximum particle size $(d_{max})$ varies significantly, ranging from 10 mm to 200 mm, as illustrated in Figure 4. Due to equipment limitations, samples with $d_{max}$ exceeding 60 mm constitute only about 10% of the 545 tests. In contrast, samples with $d_{max} = 60$ mm dominate the dataset, accounting for approximately 75% of the total. As mentioned above, we introduced the common discretization method for PSD curves, as illustrated in Figure 3. However,

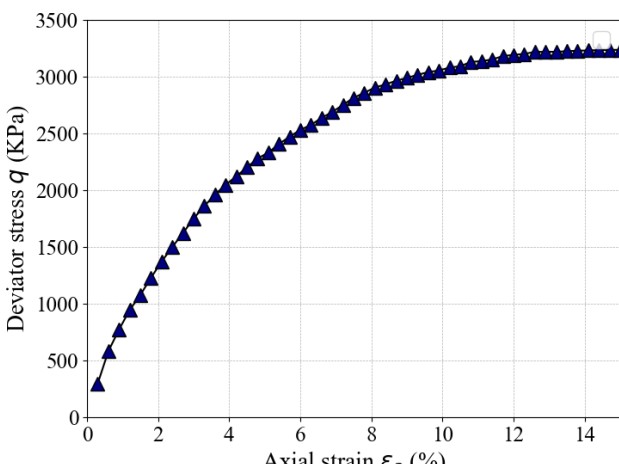

Figure 6: The deviatoric stress-axial strain $q - \varepsilon_a$ curve

Table 6: Feature list

| Features | Symbol | Unit |
|---|---|---|
| Confining Pressure | $\sigma_3$ | kPa |
| Dry Density | $\rho_d$ | g/cm$^3$ |
| Axial strain | $\varepsilon_a$ | % |
| Container height | $h$ | mm |
| Container diameter | $d$ | mm |
| Void Ratio | $e$ | % |
| The maximum particle size | $d_{\max}$ | mm |
| Particle size ratio of less than 5 mm | $P_5$ | % |
| The fraction of Particle size between 5mm and 10mm | $P_{10} - P_5$ | % |
| The fraction of Particle size between 10mm and 20mm | $P_{20} - P_{10}$ | % |
| The fraction of Particle size between 20mm and 40mm | $P_{40} - P_{20}$ | % |
| The fraction of Particle size between 40mm and 60mm | $P_{60} - P_{40}$ | % |
| The fraction of Particle size between 60mm and 80mm | $P_{80} - P_{60}$ | % |
| The fraction of Particle size between 80mm and 100mm | $P_{100} - P_{80}$ | % |
| The fraction of Particle size between 100mm and 130mm | $P_{130} - P_{100}$ | % |
| The fraction of Particle size between 130mm and 160mm | $P_{160} - P_{130}$ | % |
| Particle size ratio of more than 160 mm | $1 - P_{160}$ | % |
| Sieve aperture diameter corresponding to 20% passing mass | $D_{20}$ | mm |
| Sieve aperture diameter corresponding to 40% passing mass | $D_{40}$ | mm |
| Sieve aperture diameter corresponding to 60% passing mass | $D_{60}$ | mm |
| Sieve aperture diameter corresponding to 80% passing mass | $D_{80}$ | mm |
| Sieve aperture diameter corresponding to 100% passing mass | $D_{100}$ | mm |

a notable limitation of this method is that the number of discretization intervals varies with the maximum particle size ($d_{\max}$), leading to inconsistent feature dimensions across data samples. A simple solution is to standardize the feature dimensions by applying zero-padding, where missing features for samples with smaller particle sizes are filled with zeros. While this approach preserves PSD information for larger particle sizes, it also increases the sparsity of input features, potentially hindering the model's ability to extract meaningful patterns, especially in smaller datasets.

To address this issue, we propose a data augmentation strategy that introduces additional PSD features to enhance the input feature set. Instead of discretizing the PSD curve based on particle sizes, we adopt a mass percentage-based discretization method. In this approach, the cumulative mass percentage of coarse-grained soils passing through predefined sieve apertures is used as the discretization criterion, while the corresponding sieve aperture size serves as the feature. This method ensures that the mass percentage values consistently range from 0% to 100%, regardless of the max-

imum particle size, allowing all data samples to maintain a fixed number of PSD features based on predefined percentages such as 20%, 40%, 60%, 80%, and 100%. This process can be conceptualized as transposing the x-axis and y-axis of the original PSD curve shown in Figure 3, with the resulting features illustrated in Figure 5. The features derived from this new discretization method are then combined with those obtained through the zero-padding strategy to form the complete input feature set, as summarized in Table 6.

## A.2 B. MODEL CONFIGURATION

Table 7 provides an overview of the hyperparameter settings employed in the proposed MASR method. Furthermore, in each inner-loop iteration, 70% of the training set is dynamically selected based on the updated selection masks $\mathbf{m}$.

Table 7: MASR model configuration and hyperparameter settings

| No. | Name | Value |
|-----|------|-------|
| 1 | Learning rate | 0.0001 |
| 2 | Number of neurons in the first hidden layer | 128 |
| 3 | Number of neurons in the second hidden layer | 30 |
| 4 | Batch size | 100 |
| 5 | Activation function | PReLu |
| 6 | Optimizer | Adam |
| 7 | Penalty coefficient ($\lambda$) in IRMv1 loss ($L_{\text{IRMv1}}$) | 1 |
| 8 | Penalty coefficient ($\lambda$) in REx loss ($L_{\text{REX}}$) | 1 |
| 9 | Number of iterations (R) | 30 |

