# OpenReview forum: "Model-Agnostic Sample Reweighting for Reliable Strength Behavior Prediction of Coarse-Grained Soils Under Distribution Shifts"
_ICLR.cc/2026/Conference — Submitted to ICLR 2026_

### Official Review · Reviewer_oXTY · 2025-10-21

**Soundness:** 2
**Presentation:** 3
**Contribution:** 2
**Rating:** 2
**Confidence:** 4

**Summary:**

This paper addresses the problem of out-of-distribution (OOD) generalization for predicting the strength behavior of coarse-grained soils. It proposes a bi-level optimization framework (MASR) that integrates sample reweighting with invariant learning. While the application domain is relevant and the problem is well-motivated, the paper has significant limitations in its claimed novelty, experimental rigor, and connection to recent advances in the field.

**Strengths:**

This paper addresses a relevant and challenging problem in geotechnical engineering: reliable prediction of soil strength behavior under realistic distribution shifts. The proposed MASR framework presents a structured methodology that integrates sample reweighting with invariant learning through a bi-level optimization approach. The experimental evaluation is conducted on real-world triaxial test data, which enhances the practical relevance of the study. The method demonstrates consistent performance improvements over standard ANN and domain-specific baselines such as SNN and IRM, particularly in out-of-distribution settings. The inclusion of a sparsity constraint for sample selection also contributes to improved robustness and computational efficiency.

**Weaknesses:**

The claimed novelty of the bi-level optimization framework for sample reweighting is limited, as similar paradigms have been explored in meta-learning and robust optimization literature. The experimental comparison lacks strong and diverse OOD generalization baselines, such as Distributionally Robust Optimization (DRO) or recent methods like SWAD, which are essential for properly contextualizing the reported improvements. The reliance on a standard ANN architecture without justification or exploration of modern alternatives (e.g., Transformers) weakens the "model-agnostic" claim and raises concerns about scalability and generalizability. Additionally, the evaluation is incomplete, as performance on test samples from the dominant 50–60 mm particle size range—which constitutes 75% of the training data—is not reported, limiting the understanding of model behavior under in-distribution conditions. The dataset, while real, is relatively small, and the absence of ablation studies further restricts the interpretability of the contribution of individual MASR components.

**Questions:**

1. Limited Novelty: The core bi-level optimization for sample reweighting is not novel. The paper fails to clearly differentiate its contribution from existing meta-learning and robust optimization paradigms like ScaleBio, Meta-LoRA, ReweightOOD and so on.
2. Weak Baselines: The experimental comparison is insufficient, lacking strong OOD baselines like DRO, SWAD,  which is crucial for validating the method's claimed advantages.
3. Outdated Model Architecture: The use of a standard ANN is not justified. The performance and generalizability of the method on modern architectures (e.g., Transformers) remains unverified, weakening the "model-agnostic" claim.
4. Incomplete Evaluation: The dataset is relatively small and the performance on a critical test set from the 50-60mm range (dominant in training) is missing.

---

### Official Review · Reviewer_dFrT · 2025-11-01

**Soundness:** 2
**Presentation:** 2
**Contribution:** 2
**Rating:** 4
**Confidence:** 5

**Summary:**

The paper presents a model-agnostic sample reweighting framework that integrates sample reweighting and invariant learning via bilevel optimization, where an inner loop optimizes the model on weighted samples, and the outer loop adjusts these weights to enhance generalization under OOD conditions. This iterative mechanism enables the model to progressively identify invariant and causally relevant
features while suppressing spurious correlations. The model is applied to empirical evaluations on real-world coarse-grained soil datasets of the deviatoric stress-axial strain relationship, demonstrating that MASR outperforms baseline models in predictive accuracy under significant distribution shifts.

**Strengths:**

1 A model-agnostic sample reweighting framework that integrates sample reweighting and invariant learning is proposed to address OOD problems in the deviatoric stress-axial strain relationship of coarse-grained soils.
2 The performance demonstrated is better than other approaches reported in the literature.

**Weaknesses:**

1 In engineering problems, the dataset may not be as complete as we need. The requirement of the dataset (to cover the distributional shifts) is not discussed.
2 It is not clear how does the improvement demonstrtated in the new framework matter in addressing the OOD problem of soil mechanics.

**Questions:**

1 The accuracy of the present approach is better than others, but marginally. Can you conclude how does this minor improvement help in the problem of soils.
2 The term 'causal patterns' or 'causal relation' is mentioned in the manuscript and in references on invariant learning, but what is the 'causal relation' the problem of investigation, and how do we identify the causality should be made more rigorous.

---

### Official Review · Reviewer_nuCa · 2025-11-01

**Soundness:** 2
**Presentation:** 2
**Contribution:** 2
**Rating:** 2
**Confidence:** 4

**Summary:**

This paper proposes a new bilevel optimization framework called Model-Agnostic Sample Reweighting (MASR) for improving generalization of deep models in predicting the strength behavior of coarse-grained soils under distribution shifts. MASR integrates sample reweighting with Invariant Risk Minimization (IRM) and applies a bilevel optimization loop: the inner loop updates model parameters based on weighted samples, while the outer loop adjusts the weights and selection masks to minimize an out-of-distribution (OOD) criterion. The method is evaluated on two real-world datasets derived from triaxial compression tests, with engineered distributional shifts via varying particle size distributions. MASR is compared to ANN, IRM, and SNN baselines using metrics such as MSE, MAE, and MAPE. Results show MASR achieves consistent improvements across all settings, especially in OOD generalization.

**Strengths:**

* The paper uses 545 real triaxial compression tests with meaningful domain relevance
* MASR combines sample reweighting, IRM, and bilevel optimization in a unified pipeline, and includes a sparsity constraint.
* Evaluation across MSE, MAE, and MAPE is thorough and shows consistent improvements over baselines.

**Weaknesses:**

* IRM assumes the existence of multiple environments, but the paper never explains how these are created from the real-world dataset.
* The outer loop evaluates the OOD criterion (IRM loss) on the same training set used to compute weights. It seems to encourage overfitting sample weights to training data.
* The comparison is limited to ANN, SNN, and IRM. It omits other sample weighting and OOD generalization approaches.
* It would be better to have visualization or analysis showing correlations between high‑weight samples and physical soil characteristics.
* The authors assert that learned weights can transfer across models, but all experiments use the same ANN architecture. No cross‑model transfer is demonstrated.

**Questions:**

* How exactly are the IRM environments E defined or constructed in your dataset?
* Why is the outer‑loop OOD loss computed on the training data instead of a validation set?
* Can you visualize the learned sample weights? Do learned weights correlate with any physical soil property?
* Have you compared MASR against more advanced baselines?
* Have you tested the claimed “model‑agnostic” transfer?
* How stable are selection masks m across random seeds?
* How sensitive are your results to the choice between IRMv1 and REx?

---

### Official Review · Reviewer_Soyr · 2025-11-02

**Soundness:** 2
**Presentation:** 2
**Contribution:** 2
**Rating:** 4
**Confidence:** 4

**Summary:**

This paper proposes Model-Agnostic Sample Reweighting (MASR), a bilevel optimization framework for predicting the strength behavior of coarse-grained soils under distribution shifts. The method combines sample reweighting with invariant learning principles (IRM) to improve out-of-distribution (OOD) generalization. In the inner loop, a neural network is trained on weighted samples, while the outer loop adjusts these weights to minimize an OOD criterion. Experiments on real-world triaxial compression test data demonstrate improvements over baseline methods (ANN, SNN, IRM) across multiple evaluation metrics.

**Strengths:**

1. Addressing OOD generalization in geotechnical engineering is a relevant real-world problem.
2. Using actual triaxial compression test data rather than synthetic data adds credibility.
3. MASR shows improvements across all metrics in the experiments conducted.

**Weaknesses:**

1. Missing domain adaptation methods: No comparison with established domain adaptation techniques such as DANN (Domain-Adversarial Neural Networks), CORAL (Correlation Alignment), Deep CORAL, ADDA (Adversarial Discriminative Domain Adaptation), MMD-based methods (Maximum Mean Discrepancy).
2. Missing modern OOD methods: Group DRO (Sagawa et al., 2020), LISA (Yao et al., 2022), Fish (Shi et al., 2021) for meta-learning based domain generalization, Self-challenging methods for OOD generalization. Further, no transfer learning baselines: Simple fine-tuning or pre-training strategies not explored. This significantly undermines the claims about MASR's effectiveness, as I cannot assess whether the improvements come from the proposed method or simply from doing something beyond vanilla training.
3. The paper lacks ablation studies to systematically validate individual design choices and understand their relative contributions. The paper will benefit by providing analysis or visualization of which samples receive higher weights (which would offer crucial insights into what the model considers important).
4. The paper does not mention code availability, leaves many critical implementation details unspecified, provides no information about dataset accessibility, making it extremely difficult for others to reproduce or build upon this work.

**Questions:**

1. Why does IRM perform so poorly in your experiments? This is surprising given its theoretical foundations.
2. What is the computational overhead of MASR compared to baselines?
3. Can you provide ablation studies separating the contributions of: (a) sample reweighting, (b) sample selection via masks, (c) the specific OOD criterion choice?
4. How sensitive is the method to hyperparameters λ and the truncation length L?

---

### Meta-Review · Area_Chair_yXN3 · 2026-01-07

**Summary:**

The paper proposes Model-Agnostic Sample Reweighting (MASR), a bilevel optimization framework designed to improve the prediction of coarse-grained soil strength under distribution shifts. The method integrates sample reweighting with Invariant Risk Minimization (IRM) principles to identify stable features and suppress spurious correlations. Empirical tests were conducted on real-world triaxial compression datasets.

**Reviewer Concerns:**

Reviewers consistently highlighted the lack of comparison with modern OOD and domain adaptation baselines, limited algorithmic novelty compared to existing meta-learning paradigms, and a lack of rigorous evidence for the "model-agnostic" and "causal" claims.

**Reviewer Scores:**

Reviewers were in consensus with low scores ranging from 2 (Reject) to 4 (Marginal Reject), reflecting significant concerns regarding the paper’s technical depth and experimental scope.

---

### Decision · Program_Chairs · 2026-01-26

Reject